# Short-Period Temporal Dispersion Repolarization Markers Predict 30-Days Mortality in Decompensated Heart Failure

**DOI:** 10.3390/jcm9061879

**Published:** 2020-06-16

**Authors:** Gianfranco Piccirillo, Federica Moscucci, Gaetano Bertani, Ilaria Lospinuso, Fabiola Mastropietri, Marcella Fabietti, Teresa Sabatino, Giulia Zaccagnini, Davide Crapanzano, Ilaria Di Diego, Andrea Corrao, Pietro Rossi, Damiano Magrì

**Affiliations:** 1Anestesiologiche e Cardiovascolari, Dipartimento di Scienze Cliniche Internistiche, Policlinico Umberto I, La Sapienza University of Rome, 00185 Rome, Italy; gianfranco.piccirillo@uniroma1.it (G.P.); gaetano.bertani@uniroma1.it (G.B.); lospinuso.i@gmail.com (I.L.); fabiola.mastropietri@uniroma1.it (F.M.); marcellafabietti@gmail.com (M.F.); tere.sabatino@gmail.com (T.S.); zaccagninigiulia@gmail.com (G.Z.); davidecrap@gmail.com (D.C.); ilaria.didiego@uniroma1.it (I.D.D.); andrea.corrao@uniroma1.it (A.C.); 2Cardiology Division, Arrhythmology Unit, S. Giovanni Calibita, Isola Tiberina, 00186 Rome, Italy; rossi.ptr@gmail.com; 3Dipartimento di Medicina Clinica e Molecolare, S. Andrea Hospital, Sapienza University of Rome, 00185 Rome, Italy; damiano.magri@uniroma1.it

**Keywords:** chronic heart failure, mortality, QT, T peak–T end, QTVI, QT variability index, temporal dispersion of repolarization phase.

## Abstract

Background and Objectives: Electrocardiographic (ECG) markers of the temporal dispersion of the myocardial repolarization phase have been shown able to identify chronic heart failure (CHF) patients at high mortality risk. The present prospective single-center study sought to investigate in a well-characterized cohort of decompensated heart failure (HF) patients the ability of short-term myocardial temporal dispersion ECG variables in predicting the 30-day mortality, as well as their relationship with N-terminal Pro Brain Natriuretic Peptide (NT-proBNP) plasmatic values. Method: One hundred and thirteen subjects (male: 59, 67.8%) with decompensated CHF underwent 5 min of ECG recording, via a mobile phone. We obtained QT end (QTe), QT peak (QTp) and T peak to T end (Te) and calculated the mean, standard deviation (SD), and normalized index (VN). Results: Death occurred for 27 subjects (24%) within 30 days after admission. Most of the repolarization indexes (QTe mean (*p* < 0.05), QTe_SD_ (*p* < 0.01), QTp_SD_ (*p* < 0.05), mean Te (*p* < 0.05), Te_SD_ (*p* < 0.001) QTeVN (*p* < 0.05) and TeVN (*p* < 0.01)) were significantly higher in those CHF patients with the highest NT-proBNP (>75th percentile). In all the ECG data, only Te_SD_ was significantly and positively related to the NT-proBNP levels (r: 0.471; *p* < 0.001). In the receiver operating characteristic (ROC) analysis, the highest accuracy for 30-day mortality was found for QTe_SD_ (area under curve, AUC: 0.705, *p* < 0.01) and mean Te (AUC: 0.680, *p* < 0.01), whereas for the NT-proBNP values higher than the 75th percentile, the highest accuracy was found for Te_SD_ (AUC: 0.736, *p* < 0.001) and QTe_SD_ (AUC: 0.696, *p* < 0.01). Conclusion: Both mean Te and Te_SD_ could be considered as reliable markers of worsening HF and of 30-day mortality. Although larger and possibly interventional studies are needed to confirm our preliminary finding, these non-invasive and transmissible ECG parameters could be helpful in the remote monitoring of advanced HF patients and, possibly, in their clinical management. (ClinicalTrials.gov number, NCT04127162).

## 1. Introduction

The markers of the temporal dispersion of the myocardial repolarization phase have been shown to able to identify those heart failure (HF) patients at high risk, either in terms of all-cause mortality [1,2] or sudden cardiac death due to malignant ventricular arrhythmias [2,3,4]. Indeed, it is widely accepted that myocardial repolarization might suffer from a number of possible conditions due to the complex interplay between ionic membrane channels, membranes’ transporter mechanisms and many cardiac and extra-cardiac multi-organ regulatory systems.

Of note, besides the direct myocardial structural damages (i.e., ischemia, necrosis, fibrosis, hypertrophy, disarray, etc.) [5,6], drugs [7,8,9], aging [9,10,11,12], mental stress [13,14,15] and neurohumoral activation [4,10,12,16,17,18,19,20,21], as well as the immune–inflammatory response [22,23] are all factors able to worsen the myocardial repolarization phase. Thus, due to the abovementioned high susceptibility to different possible stressors, there is growing interest in the analysis of myocardial repolarization phase dispersion as a possible marker of global cardiovascular risk rather than a mere non-invasive electrophysiological marker of electrical instability. Particularly, changes in the electrocardiographic (ECG)-derived variables associated with the myocardial repolarization phase might be extremely useful as immediate markers of complex structural and molecular mechanisms leading to clinical decompensation, as well as possibly lethal arrhythmic complications [24]. These non-invasive and relatively easy to obtain ECG-derived data, in a setting of increasingly ageing population with many comorbidities, could represent a useful tool in the management of chronic patients in the contexts of non-intensive care, institutionalization or even at home. In such a context, the short-term evaluation of the interval between the T peak and T end (Te) segment represents one of the most promising approaches in myocardial repolarization dispersion analysis [25]. 

One of the clinical settings which could benefit from this approach is undoubtedly represented by patients with chronic heart failure (CHF) syndrome, which is the HF decompensation, a frequent event characterized by a high risk of mortality and morbidity. Accordingly, a prompt detection of a worsening myocardial repolarization dispersion might enable us to more strictly manage the patients by modifying their therapeutic regimen or by reducing the intervals between their medical examinations.

Therefore, the present prospective single-center study sought to investigate in a well-characterized cohort of decompensated HF patients the ability of a number of short-term myocardial temporal dispersion ECG variables and, particularly of those derived from the analysis of the Te interval, of predicting the 30-day cardiovascular and total mortality. Furthermore, due to the well-known role of the plasmatic levels of N-terminal Pro Brain Natriuretic Peptide (NT-proBNP) in identifying those HF patients at high risk, we also evaluated the ability of the temporal dispersion of myocardial repolarization to identify those decompensated HF patients with the highest NT-proBNP values defined according to an arbitrary cutoff value corresponding to the 75th percentile.

## 2. Methods

### 2.1. Patients and Protocol

We enrolled a total 117 consecutive HF patients admitted to our Geriatric Medicine Department from January 2019 to January 2020 due to a decompensated CHF condition, the latter defined according to the 2016 European Society of Cardiology HF guidelines [26]. All patients had stable previous clinical conditions at home with NYHA class II-III and all of them were in the fourth NYHA functional class at the time of enrollment.

At the study run-in, each patient underwent a full clinical history, physical examination, standard ECG and standard transthoracic echocardiogram. Furthermore, a 5-min single lead (II lead) ECG recording (MiocardioEvent^TM^, Rome, Italy) and a blood sample for NT-proBNP dosage (Alere Triage Analyzer, Alere, San Diego, CA, USA) and other serum variables were obtained. The Cockcroft–Gault formula was used to assess the creatinine clearance.

All patients provided informed consent for the use of their records for research purposes and the study was in accordance with good clinical practice and the principles of the Declaration of Helsinki for clinical research involving human patients. The study underwent Ethical Committee of Policlinico Umberto I approbation. The ClinicalTrials.gov number is NCT04127162.

### 2.2. Offline Data Analysis

The 5-min single lead (D II) ECG signals (Miocardio Event™, Rome, Italy) were acquired, digitized at a sampling frequency of 500 Hz and wirelessly transmitted to a cloud platform for data storage via a mobile phone. Subsequently, all digitized signal recordings were downloaded by the cloud platform and analyzed by a single physician (G.P.) blinded to the subjects’ circumstances. Thereafter, the following ECG intervals from the respective time series were obtained: RR, QT end (QTe), QT peak (QTp) and T peak to T end (Te). Briefly, the QTe was obtained by measuring the interval from the onset of the Q-wave to the T-wave end; the QTp was obtained by measuring the interval from the onset of the Q-wave to the T-wave peak; the Te was obtained from the T-wave peak to the T-wave end. To identify the abovementioned ECG segments, software originally proposed by Berger [27] and validated in other subsequent studies was used [2,9,10,11,12,13,16,17,18,19,28,29,30,31]. We therefore calculated the mean, variance and standard deviation (SD) values for each of these repolarization phase intervals and, finally, we calculated the variance normalized for the mean (VN), according to the following Formulas (1)–(3) [2,4,9,10,11,12,13,16,17,18,19,28,29,30,31]:QTeVN = QTe variance/(QTemean)^2^(1)
QTpVN = QTp variance/(QTp mean)^2^(2)
TeVI = Te variance/(Te mean)^2^(3)

Software for data analysis was designed and produced by our research group with the LabView program (National Instruments, Austin, TX, USA).

### 2.3. Statistical Analysis

All variables with a normal distribution were expressed as mean ± standard deviation, except for non-normally distributed variables, such as as median and inter-quartile range (i.r.). Two distinct subgroups (Group 1 and 2) were identified according to an arbitrarily defined value of NT-proBNP corresponding to the 75th percentile value obtained in the overall study sample (i.e., <75th and ≥75th percentile).

A one-way analysis of variance (ANOVA)and Bonferroni test were used to compare data for the normally distributed variables; Kruskal–Wallis and Mann–Whitney tests were used to compare non-normally distributed variables (as evaluated by a Kolmogorov–Smirnov test). Univariable Cox proportional-hazards regression analysis was used to test the association between continuous and dichotomized variables with 30-day mortality. The temporal myocardial repolarization dispersion variables with the highest χ^2^ were used for Kaplan–Meier curves [2,32].

Receiver operating characteristic (ROC) curves were used to determine the sensitivity and specificity of the studied parameters predictive of mortality and areas under curves (AUCs) and 95% confidence intervals (CIs) were calculated to compare the diagnostic accuracy.

All data were evaluated by use of database SPSS-PC+ (SPSS-PC+ Inc, Chicago, IL, USA). A *p*-value less than or equal to 0.05 were considered statistically significant.

## 3. Results

Starting from 117 eligible patients, four patients were excluded because of poor quality ECGs. Accordingly, a total of 113 decompensated HF patients were analyzed and prospectively followed. The 75th percentile value of NT-proBNP in the overall population was equal to 6.660 pg/mL and, accordingly, two distinct subgroups were defined (Group 1 and Group 2).

Table 1 shows the detailed comparison between the two study groups. Group 1 consisted of 85 patients, while Group 2 consisted of 28 subjects (Table 1). Except for the left ventricular ejection fraction (LVEF) and, obviously, the NT-proBNP values, the two groups showed similar clinical characteristics. At the study run-in, all patients were in NYHA functional class IV.

As reported in detail in Table 2, most of the ECG-derived data dealing with the temporal dispersion of the repolarization data were significantly lower in Group 1 than in the counterpart. Of note, with respect to specific patients’ subsets, those with atrial fibrillation showed higher values of QTp_SD_ (*p* < 0.01), Te_SD_ (*p* < 0.05), QTpVN (*p* < 0.05) and TeVN (*p* < 0.05), but no differences with respect to QTe, QTp and Te means. Conversely, in the 26 patients with a paced rhythm, a significantly longer QTe (479 ± 58 vs. 424 ± 58 ms, *p* < 0.001), QTp (366 ± 76 vs. 322 ± 55 ms, *p* < 0.01) and Te (112 ± 24 vs. 102 ± 22 ms, *p* < 0.05) were observed.

A positive relationship was found between NT-proBNP and Te_SD_, the latter expressed on a logarithmic scale (r: 0.471; *p* < 0.001) (Figure 1).

A total of 27 (24%) patients died within 30 days: 18 patients died from respiratory failure, six from terminal heart failure, two from fatal acute myocardial infarction and one patient from arrhythmic sudden cardiac death (sustained ventricular tachycardia and ventricular fibrillation). Thus, we classified 27 patients as total mortality and nine as cardiovascular mortality. Specifically, 17 patients among Group 1 died (20%) and ten (36%) in Group 2.

Comparing surviving and deceased patients (Figure 2A,B), mean Te (*p* < 0.001), QTeSD (*p* < 0.05) and Te SD (*p* < 0,05) were shown to be highly predictive forpoor prognosis.

We also obtained ROC curves for the NT-proBNP values higher than the 75th percentile and the following were the ECG variables with the best accuracy: Te_SD_ (AUC: 0.736, *p* < 0.001), QTe_SD_ (AUC: 0.696, *p* < 0.01), TeVN (AUC: 0.674, *p* < 0.01), QTp_SD_ (AUC: 0.655, *p* < 0.05), QTeVN (AUC: 0.650, *p* < 0.05) and mean QTe (AUC: 0.639, *p* < 0.05) (Figure 3B).

The survival curve constructed for the 75th percentile value of mean Te exhibited a good significance for total mortality (Figure 4A) and cardiovascular mortality (Figure 4B).

## 4. Discussion

The major finding of the present study was the significant relationship between some short-period ECG markers of the repolarization phase with worsening HF, as well as 30-day mortality. Particularly, the TE_SD_ was linearly related to the NTproBNP plasma level and the simple mean Te values was an independent predictor of both all-cause and cardiovascular mortality in our study sample. 

Our data showed a 30-day mortality of 24%, higher than that reported in another similar study, where a mortality risk ranging from 2.1 to 21.9% was found [32]. However, it should be noted that the abovementioned study evaluated patients significantly younger (<75 years old) than those we enrolled (i.e., mean age of 83 years old). 

During CHF, ventricular repolarization is deeply involved, and the more advanced this clinical syndrome is, the longer the duration and the more the temporal homogeneity of repolarization is affected [4,16,25]. The pathological basis of these changes includes the morphological remodeling of the histological substrate (myocyte hypertrophy, disarray, fibrosis, etc.), especially ion channel remodeling. In fact, in the last twenty years, many authors highlighted potassium (*I*K_to_, *I*K_s_, *I*K_r_ and *I*K1), sodium channels (*I*Na) and calcium handling alterations. In particular, CHF was associated with the downregulation of *I*K_to_, *I*K_s_, *I*K_r_ and *I*K1; on the contrary, *I*Na inactivation occurs later in comparison with normal subjects and finally, the release and storage of calcium in the sarcoplasmic reticulum were found to be abnormal [33]. The results of these alterations, as already mentioned above, is an increase in the duration and the temporal inhomogeneity of repolarization.

Although Te was studied as non-invasive marker of sudden cardiac risk, a recent meta-analysis [34] of more than 150,000 subjects evidenced that it might act, not only as a marker of ventricular arrhythmic or sudden cardiac death risk, but also as a risk marker of total and cardiac death. Our data are consistent with this meta-analysis, the most accurate mean Te cut-off value for mortality risk in our study was higher than 13 ms, in comparison with the one found in the meta-analysis. However, in our opinion, the novel and interesting datum was that a dynamic parameter, Te_SD_, was associated with a well-known biomarker of HF severity, NT-proBNP, thus enabling us to speculate that this dynamic ECG repolarization marker could be used for the remote monitoring of CHF patients. In fact, this non-invasive, repeatable, inexpensive and transmissible ECG parameter could be helpful in monitoring those HF patients at high risk of worsening or mortality [35].

Up to now, the pathophysiologic basis of the relation between NT-proBNP and Te_SD_ remains to be clarified in decompensated CHF, but it is likely that both markers suffer from neuro-humoral activation [36] in terms of sympathetic and renin–angiotensin–aldosterone system overstimulation [16,37]. Accordingly, it was well known that catecholamines are able to decrease the current density of *I*K_to_, *I*K_1_, *I*K_s_ and L-type Ca^2+^ channels and to upregulate T-type Ca^2+^ channels [38,39] and, contextually, it has been reported that angiotesin II decreases the current density of *I*K_1_, *I*K_to_ and *I*K_ur_ [40,41]. Aldosterone also induces an upregulation of L-type Ca^2+^ and Na^+^ channels [42].

Finally, we would remark that, although most of the decompensated HF patients were not in sinus rhythm (atrial fibrillation, frequent premature ventricular or supraventricular contractions, electrostimulated subjects, etc.), the non-invasive ECG markers we found predictive in our study, namely Te and Te_SD_, do not need a sinus rhythm condition to be calculated. Mean Te and Te_SD_ provided reliable data in this subject category. 

## 5. Conclusions

In conclusion, mean Te and Te_SD_ could be considered as reliable markers of worsening HF and of 30-day mortality. Although the small number of patients enrolled and the observational single-center nature of the study, our data might be useful from a clinical viewpoint. Indeed, having a non-invasive marker, predictive of clinical worsening and mortality in a complex syndrome such as the advanced HF, could guarantee prompt intervention by the clinician and immediate and targeted therapeutic choices. The small number of patients did not allow a multivariate Cox analysis. Obviously, larger and possibly interventional studies are needed to confirm our preliminary finding.

## Figures and Tables

**Figure 1 jcm-09-01879-f001:**
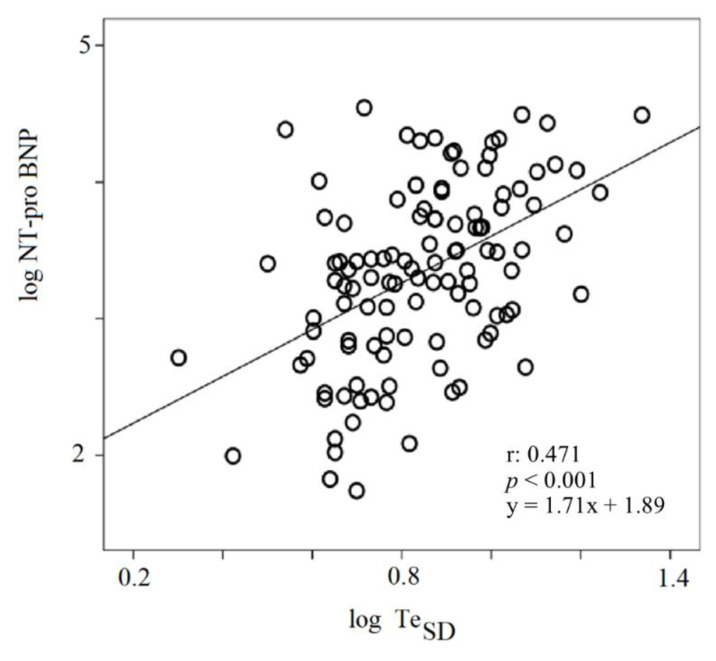
Linear regression between log Te_SD_ and NT-proBNP. NT-proBNP: N-terminal Pro Brain Natriuretic Peptide; SD: standard deviation.

**Figure 2 jcm-09-01879-f002:**
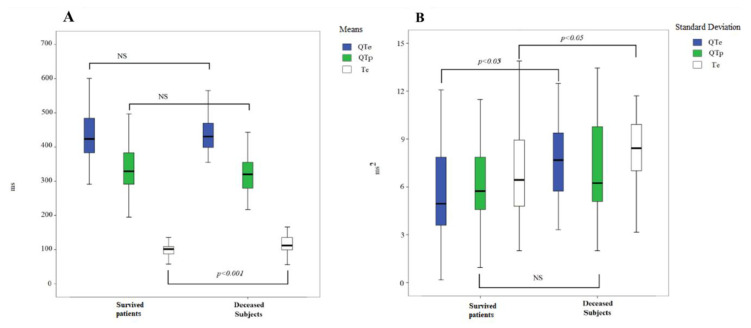
(**A**) QTe, QTp and Te means in the cardiovascular and non-cardiovascular deceased subjects. In the box plots, the central line represents the median distribution. Each box spans from the 25th to 75th percentile points, and error bars extend from the 10th to 90th percentile points. (**B**) QTe, QTp and Te standard deviations (SD) in the cardiovascular and non-cardiovascular deceased subjects. In the box plots, the central line represents the median distribution. Each box spans from the 25th to 75th percentile points, and error bars extend from the 10th to 90th percentile points.In the ROC analysis, the highest accuracy for 30-day mortality was found for the following variables: QTe_SD_ (AUC: 0.705, *p* < 0.01), mean Te (AUC: 0.680, *p* < 0.01), QTeVN (AUC: 0.686, *p* < 0.01) and Te_SD_ (AUC: 0.648, *p* < 0.05) (Figure 3A). ROC, receiver operating characteristic; AUC, area under curve; NS, not significant. QTe, QTp, Te, QTeVN (explained in the text).

**Figure 3 jcm-09-01879-f003:**
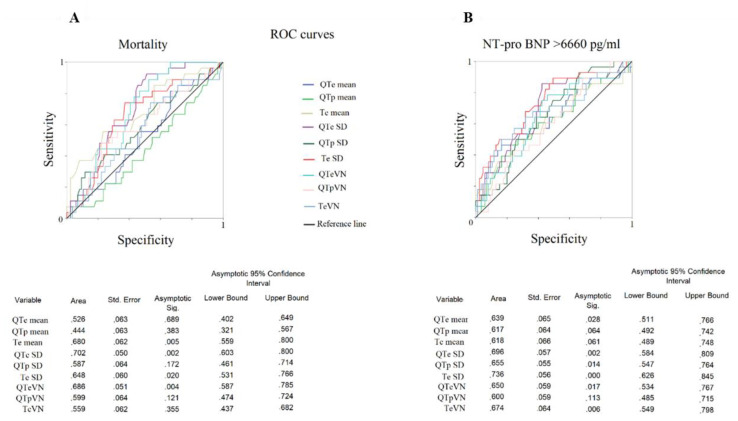
(**A**,**B**) ROC curve of statistically significant examined variables. Sensitivity–specificity of different variables for mortality (**A**) and higher NT-proBNP levels (≥75th percentile) (**B**). Univariable Cox regression analysis reported a significant relationship between 30-day mortality and QTe_SD_ (hazard ratio: 1.10, 95% confidence limit: 1.03–1.18, *p* < 0.05), mean Te (hazard ratio 1.02, 95% confidence limit: 1.01–1.04, *p* < 0.001) and Te_SD_ (hazard ratio 1.12, 95% confidence limit: 1.01–1.23, *p* < 0.001) (Table 3). All number value in the table of the figure should be intended as follow: .526 means 0.526.

**Figure 4 jcm-09-01879-f004:**
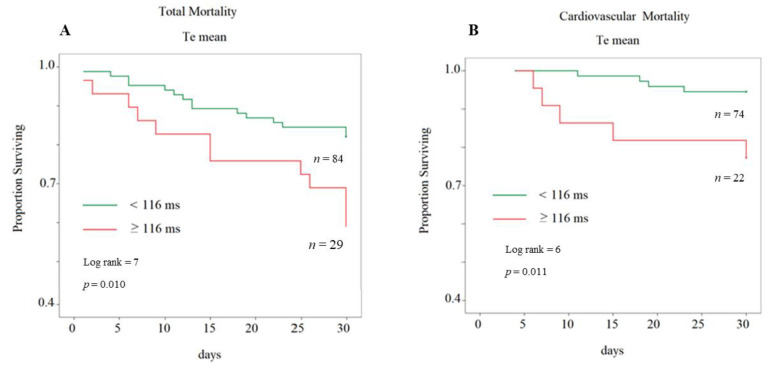
(**A**) Kaplan–Meier survival curve for total mortality by subdividing the study subjects with a cutoff at the 75th percentile of mean Te. (**B**) Kaplan–Meier survival curve for cardiovascular mortality by subdividing the study subjects with a cutoff at the 75th percentile of mean Te.

**Table 1 jcm-09-01879-t001:** General characteristics of the study subjects.

	<75th	≥75th	
	of NT-proBNP		
	Group 1	Group 2	
	*n*: 85	*n*: 28	*p*
Age, years	82 ± 10	85 ± 11	ns
Male/Female, *n*	44/41	15/13	ns
BMI, kg/m^2^	26 ± 5	26 ± 5	ns
SBP, mm Hg	127 ± 21	124 ± 20	ns
DBP, mm Hg	70 ± 11	65 ± 11	ns
Heart Rate, b/m	74 ± 14	76 ± 14	ns
Left Ventricular Ejection Fraction, %	46 ± 9	35 ± 9	<0.001
NT-proBNP, pg/mL	1530 (2225)	13,100 (11,780)	<0.001
Serum Potassium, mmol/L	4.1 ± 0.5	4.3 ± 0.6	ns
Serum Calcium, mmol/L	2.2 ± 0.2	2.1 ± 0.4	ns
Creatinine Clearance, mL/m	49 ± 27	38 ± 30	ns
CHF with Depressed Systolic Function, *n* (%)	44 (52)	24 (86)	ns
CHF with Preserved Systolic Function, *n* (%)	59 (69)	13 (46)	ns
Hypertension, *n* (%)	64 (75)	19 (68)	ns
Hypercholesterolemia, *n* (%)	33 (39)	17 (61)	ns
Diabetes, *n* (%)	31 (36)	13 (46)	ns
Known Myocardial Ischemia History, *n* (%)	24 (28)	10 (36)	ns
Valve Diseases	12 (14)	4 (14)	ns
Premature Supraventricular Complexes, *n* (%)	12 (14)	5 (18)	ns
Premature Ventricular Complexes, *n* (%)	22 (26)	6 (21)	ns
Permanent Atrial fibrillation, *n* (%)	29 (34)	9 (32)	ns
Left Bundle Branch Block, *n* (%)	11 (13)	7 (25)	ns
Right Bundle Branch Block, *n* (%)	15 (18)	7 (25)	ns
Pacemaker-ICD, *n* (%)	16 (19)	10 (36)	ns
β-blockers, *n* (%)	49 (58)	24 (86)	ns
Furosemide, *n* (%)	61 (72)	26 (93)	ns
ACEi/ARB	36 (42)	9 (32)	ns
Aldosterone antagonists, *n* (%)	11 (13)	3 (11)	ns
Potassium, *n* (%)	5 (6)	2 (7)	ns
Nitrates, *n* (%)	10 (12)	4 (14)	ns
Ivabradine, *n* (%)	3 (4)	2 (7)	ns
Digoxin, *n* (%)	4 (5)	2 (7)	ns
Statins, *n* (%)	22 (26)	12 (43)	ns
Antiplatelet drugs, *n* (%)	32 (38)	6 (21)	ns
Oral Anticoagulants, *n* (%)	21 (25)	9 (32)	ns
Diltiazem or Verapamil, *n* (%)	6 (7)	0 (0)	ns
Dihydropyridine Calcium channel blockers, *n* (%)	13 (15)	2 (7)	ns
Propafenone, *n* (%)	1 (1)	0 (0)	ns
Amiodarone, *n* (%)	1 (1)	4 (14)	<0.05
Ranolazine, *n* (%)	5 (6)	0 (0)	ns
Sacubitril/Valsartan, *n* (%)	1 (1)	0 (0)	ns

BMI: body mass index, SBP: systolic blood pressure, DBP: diastolic blood pressure; CHF depressed ejection fraction: ejection fraction <50%; CHF preserved ejection fraction: ejection fraction >50%; ACEi: angiotensin converting enzyme inhibitors; ARB: angiotensin receptors blockers; NT-proBNP: N-terminal Pro Brain Natriuretic Peptide; SD: standard deviation; ICD: implantable cardioverter defibrillator. Data are expressed as mean ± SD, or median (interquartile range), or number of patients (%).

**Table 2 jcm-09-01879-t002:** Short period repolarization temporal dispersion variables in study patients.

	<75th	≥75th	
	of NT-proBNP		
	Group 1	Group 2	
	*n*: 85	*n*: 28	*p*
QTe mean, ms	428 ± 62	463 ± 85	0.020
QTe SD, ms	5 (4)	8 (5)	0.002
QTe SD, log ms	0.69 ± 0.29	0.87 ± 0.24	0.003
QTp mean, ms	326 ± 60	351 ± 69	ns
QTp SD, ms	5 (3)	7 (4)	0.014
QTp SD, log ms	0.74 ± 0.20	0.85 ± 0.16	0.009
Te mean, ms	101 ± 18	113 ± 32	0.023
Te SD, ms	6 (4)	9 (5)	<0.001
Te SD, ms	0.80 ± 0.16	0.96 ± 0.18	<0.001
QTeVN	0.15 (0.32)	0.30 (0.41)	0.017
QTpVN	0.29 (0.49)	0.36 (0.56)	ns
TeVN	4 (4)	9 (11)	0.006

QTe, QTp, QTeVN, QTpVN, TeVN (explained in the text). Data are expressed as mean ± SD, or median (interquartile range), or number of patients (%).

**Table 3 jcm-09-01879-t003:** Prediction of mortality in study patients by Cox regression (continuous variables).

Variables	Χ^2^	Univariable Analysis Hazard Ratio (95% CI)	*p-*Values
QTe SD	7.60	1.10 (1.03–1.18)	0.006
Mean Te	12.18	1.02 (1.01–1.04)	< 0.001
Te SD	4.65	1.12 (1.01–1.23)	0.026

Data are presented as: hazard ratio (95% confidence limit).

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
