# Peer review of "Short-Period Temporal Dispersion Repolarization Markers Predict 30-Days Mortality in Decompensated Heart Failure"

_jcm, 2020, doi:10.3390/jcm9061879_

Round 1
Reviewer 1 Report
I want to thank the authors for the changes made based on my suggestions. I have no further queries regarding this article.
Regards.
Reviewer 2 Report
Authors have worked a lot on the manuscript on this round. The manuscript is much clearer now and well presented and I have no further comments.
This manuscript is a resubmission of an earlier submission. The following is a list of the peer review reports and author responses from that submission.
Round 1
Reviewer 1 Report
I have read with interest the article by G. Piccirillo et al. in which prognosis value of short-period temporal dispersion repolarization markers were evaluated in 113 patients after an episode of acute decompensated heart failure (HF). I find very interesting to find tools that allow us to predict the prognosis of patients with HF from the ECG, since it could complement the use we are currently giving to telemedicine for the follow-up of these patients, however, I consider that this article has several limitations:
The included patients are in a very advanced stage of the disease: NYHA functional class IV, mean age 83 years old, 30-day mortality> 20% ... Does not seem to be a population in which it is useful to know parameters that help us to predict mortality, rather they are patients in whom we should focus on palliative care and end-of-life.
117 patients were included in NYHA functional class IV. Methods section does not explain when NYHA was evaluated. Is was at the time of admission? Or it was about the functional class prior to decompensation? The same happens regarding levels of NTproBNP, it is about the blood test in the emergency department, on admission or at discharge?
The criteria used to define HF decompensation (patients with at least one symptom or sign compatible with a decompensation and a previous documented history of CHF) do not correspond to the current clinical practice guidelines.
I consider that Table 1 should include not only the characteristics of the total cohort but also the differences between the 3 groups according NTproBNP. Differences in NTproBNP levels could be due to different renal function, age, or BMI rather than due to the severity of HF.
It is a study about repolarization, it should also include information on analytical parameters such as potassium or calcium.
Table 1: We do not know when the heart rate or blood preassure was evaluated.
Mortality analyses are made on total mortality when more than half of the deaths (18 of 27) were of respiratory cause. What role can repolarization have in non-cardiovascular mortality? I believe that the analyzes should be done regarding cardiovascular mortality (HF, sudden death etc.)
Why were the NTproBNP groups made in this way? Percentile≤50th, 50-75 and ≥75th. Why not separate it by tertiles or in any other more homogeneous way?
Results page 5 line 135: You are comparing groups 1 and 3 but referring to table 2 in which the p-value is the result of comparing the 3 groups with each other, that is not correct.
In the entire discussion section there is no a single word dedicated to the limitations of the study.
Minor changes:
Abstract, line 20: 113 subjects (M / F: 60/53) please use percentages
Abstract, line 27: “were increased in CHF subjects with higher level (≤75th percentile)…” change to (≥75th percentile)
Methods, page 2, line 60: "patients admitted to our department from January ??? To January 2020."
Table 1: again M / F: please use n and %
Table 2: ACE / Sartans change to ACEI / ARB
Table 2: Sacubitril: The correct name is sacubitril / valsartan
Author Response
#Reply to Reviewer 1
We would thank the reviewer for his/her encouraging comments and we proceeded throughout a new revision with his/her useful remarks.
Poit-by-point answer:
-The included patients are in a very advanced stage of the disease: NYHA functional class IV, mean age 83 years old, 30-day mortality> 20% ... Does not seem to be a population in which it is useful to know parameters that help us to predict mortality, rather they are patients in whom we should focus on palliative care and end-of-life.
Answer: We thank for the observation, which gives us the opportunity to explain some things: our department deals mainly with geriatric patients, who have a high degree of polypathology and comorbidity, in this case the NYHA IV class has to be considered the consequence of an acute heart failure episode and not the usual clinical status of patients. We agree in considering this category of patients at high risk of mortality; however, we are not a Hospice, we do not palliation. In Italy, we have many elderly patients and understanding which of them are at higher risk of developing complications in the short and medium term during chronic heart disease, it seems to us a crucial aspect for the organization of medicine outside hospitals and in daily life of patients monitoring.
-117 patients were included in NYHA functional class IV. Methods section does not explain when NYHA was evaluated. Is was at the time of admission? Or it was about the functional class prior to decompensation? The same happens regarding levels of NTproBNP, it is about the blood test in the emergency department, on admission or at discharge?
Answer: We have slightly changed the introduction of the Methods to better understand that the clinical, electrocardiographic, instrumental and biohumoral evaluation of the patients took place at the time of enrollment, which coincided with the entry into our ward for some patients or at later times if heart failure occurred during hospitalization. None of these patients was previously in Class NYHA IV previously, and decompensation was often the reason for entering the hospital.
-The criteria used to define HF decompensation (patients with at least one symptom or sign compatible with a decompensation and a previous documented history of CHF) do not correspond to the current clinical practice guidelines.
Answer: We would thank very much the reviewer for this comment, which gave us the opportunity to improve the sentence. We agree that in the sentence of the previous version the concept was poorly expressed: our aim was to communicate the inclusion of both subjects with a reduced ejection fraction, both preserved. In fact, diastolic heart failure, even of an advanced degree, is rather typical of the elderly (hypertensive, diabetic, dyslipidemic, etc.). Also, this type of heart failure, as is known, can undergo clinical exacerbations with the increase of specific biohumoral markers. Therefore, it seemed appropriate to include this category of patients also to demonstrate the reliability of the electrocardiographic markers in these patients, who are the major part of elderly patients.
-I consider that Table 1 should include not only the characteristics of the total cohort but also the differences between the 3 groups according NTproBNP. Differences in NTproBNP levels could be due to different renal function, age, or BMI rather than due to the severity of HF.
Answer: thank you for the suggestion, we have completely changed table 1 in order to include new data.
-It is a study about repolarization, it should also include information on analytical parameters such as potassium or calcium.
Answer: this is an excellent observation, we have previously examined these data, but thus they were not statistically relevant, we have missed them in the previous version. Now, you can find them in the table.
-Table 1: We do not know when the heart rate or blood pressure was evaluated.
Answer: we have amended.
-Mortality analyses are made on total mortality when more than half of the deaths (18 of 27) were of respiratory cause. What role can repolarization have in non-cardiovascular mortality? I believe that the analyzes should be done regarding cardiovascular mortality (HF, sudden death etc.)
Answer: thank you for this comment. We have changed figures 3 and 4 analyzing total and cardiovascular mortality. The fact that we have already observed that these ECG markers could be predictive for total and not only for cardiovascular mortality could be an advantage from the clinical point of view, giving to these acquisitions more usefulness.
Why were the NTproBNP groups made in this way? Percentile≤50th, 50-75 and ≥75th. Why not separate it by tertiles or in any other more homogeneous way?
Answer: following the suggestion of referee n. 2 we have changed the subgroups analyzed and we obtained 2 groups using 75th percentile cut-off.
Results page 5 line 135: You are comparing groups 1 and 3 but referring to table 2 in which the p-value is the result of comparing the 3 groups with each other, that is not correct.
Answer: we have completely changed tables and text referring to these data.
In the entire discussion section there is no a single word dedicated to the limitations of the study.
-We have added a sentence to highlight limitations.
Finally, we are in debt with you.
Reviewer 2 Report
Summary
In this paper, authors derive a set of markers of dispersion of repolarization from 5 min ECG at rest. The endpoints are total mortality and NT-pro BNP, which is a known biomarker of CHF. They find the mean and SD Tpe are significantly higher in the group of patients who suffered a cardiovascular death, from those who didn’t, and claim these indices could help non-invasive monitoring of risk. The study is important, but it is not clear what is the novelty, and the clarity of the manuscript could be improved.
Major comments:
- Abstract: QTeV and TeV are not defined. It needs to be clear in Methods that you tested for two endpoints: mortality and 75th percentile of NT-pro BNP. This also applies to the main text.
- Introduction: I found it very short and not very useful. You should motivate your work based on the niche/gap of previous studies. What are your hypotheses? In addition, the objectives of this work are not clear. Am I right in thinking you have two, (1) quantify the association of repolarization dispersion markers with mortality, and (2) quantify their association with NT-pro BNP? This is not clear, and the endpoints of the study need to be very clear.
- Methods – Offline data analysis: You mention two ECG datasets were acquired: a standard ECG and a lead II ECG. Which one did you analyse and why? Also, was QTe measured from Q-wave onset or Q-wave peak? Please clarify.
- Methods – Statistical analysis: You analyse data in three subgroups based on the 50th and 75th percentiles of NT-pro BNP. Why? What is the motivation for this? Is there any previous work that supports it? You are later using only two groups (those with NT-pro BNP < 75th versus those with NT-pro BNP > 75th). I suggest using only these two groups. Finally, authors perform multiple statistical tests, so they shoud apply Bonferroni correction.
- Results – Tables 1 and 2: Please add the same data for four additional columns. The first two would be those who died (cases) and those who didn’t (controls), and the last two would be those with NT-pro BNP < 75th and those with NT-pro BNP > 75th.
- Results – Figure 1: Why are you analysing the correlation between NT-pro BNP and TeSD in the logarithmic scale? Also, these results should be reported before Table 3, and not only for Tpe, but also for all the other markers of dispersion of repolarization.
- Results – Figure 2: Why are you only reporting the KM curves for these indices? How are the curves for the other ones, and what is the motivation for not reporting them?
- Results – Table 4: What are the units of the hazard ratios? In addition, why does the multivariate model only adjust for age, BMI, NR-pro BNP, EF, SBP and DBP? I suggest you include in the table all variables from Table 1. Then you add all those that are significant in the univariate model to the multivariate model.
- Results – Figure 3: ROC curves (and correlation plots) should be reported before any survival analysis.
- Discussion – What does the second paragraph contribute? Next, how does the third paragraph link to your findings? In addition, how do your results change when using the cut-off from the meta-analysis (reference [30]), instead of your 75th percentile? Finally, you need to highlight again how your work is different from previous studies? What does it add?
- English needs to be checked and improved across the manuscript.
Minor comments:
- In section 2.1, when did the enrolment start? It only says “January” at the moment.
- Mention to Table 2 on page 5 should be to Table 3 instead?
Author Response
#Reply to Reviewer 2
We would thank the reviewer for his/her encouraging comments and we proceeded throughout a new revision with his/her useful remarks.
Summary
In this paper, authors derive a set of markers of dispersion of repolarization from 5 min ECG at rest. The endpoints are total mortality and NT-pro BNP, which is a known biomarker of CHF. They find the mean and SD Tpe are significantly higher in the group of patients who suffered a cardiovascular death, from those who didn’t, and claim these indices could help non-invasive monitoring of risk. The study is important, but it is not clear what is the novelty, and the clarity of the manuscript could be improved.
Major comments:
- Abstract: QTeV and TeV are not defined. It needs to be clear in Methods that you tested for two endpoints: mortality and 75th percentile of NT-pro BNP. This also applies to the main text.
Answer:thank you for this observation, we have highlighted these points.
- Introduction: I found it very short and not very useful. You should motivate your work based on the niche/gap of previous studies. What are your hypotheses? In addition, the objectives of this work are not clear. Am I right in thinking you have two, (1) quantify the association of repolarization dispersion markers with mortality, and (2) quantify their association with NT-pro BNP? This is not clear, and the endpoints of the study need to be very clear.
Answer: Thanks to your analysis, we have improved this point.
- Methods – Offline data analysis: You mention two ECG datasets were acquired: a standard ECG and a lead II ECG. Which one did you analyse and why? Also, was QTe measured from Q-wave onset or Q-wave peak? Please clarify.
Answer: it is true, so we have better clarified this point
- Methods – Statistical analysis: You analyse data in three subgroups based on the 50th and 75th percentiles of NT-pro BNP. Why? What is the motivation for this? Is there any previous work that supports it? You are later using only two groups (those with NT-pro BNP < 75thversus those with NT-pro BNP > 75th). I suggest using only these two groups. Finally, authors perform multiple statistical tests, so they shoud apply Bonferroni correction.
Answer: thank you for your observation. Following your suggestion, we have revised the statistical analysis, in order to obtain two subgroups, using NT pro BPN 75th percentile cut-off.
- Results – Tables 1 and 2: Please add the same data for four additional columns. The first two would be those who died (cases) and those who didn’t (controls), and the last two would be those with NT-pro BNP < 75th and those with NT-pro BNP > 75th.
Answer: case-control studies are retrospective; our study was planned as a prospective observational study, so we cannot analyze data in these ways. In particular, died patients could not be considered “cases” as the not died could not be considered “controls”. So, we have preferred report data in this way.
- Results – Figure 1: Why are you analysing the correlation between NT-pro BNP and TeSD in the logarithmic scale? Also, these results should be reported before Table 3, and not only for Tpe, but also for all the other markers of dispersion of repolarization.
Answer: thank you for the suggestion; we have modified this point.
- Results – Figure 2: Why are you only reporting the KM curves for these indices? How are the curves for the other ones, and what is the motivation for not reporting them?
Answer: we have better clarified our choice in the statistical paragraph, in which we have reported 2 specific references reporting the same analysis.
- Results – Table 4: What are the units of the hazard ratios? In addition, why does the multivariate model only adjust for age, BMI, NR-pro BNP, EF, SBP and DBP? I suggest you include in the table all variables from Table 1. Then you add all those that are significant in the univariate model to the multivariate model.
Answer: we have conducted this analysis without any significant statistical results, so we decided, to not include these data in the table, reporting the only ones that were of some interest for the reader. Our interest is focuses on ECG markers and not on clinical variables, which have been already studied in many other studies in the past.
- Results – Figure 3: ROC curves (and correlation plots) should be reported before any survival analysis.
Answer: thank you for the suggestion, we have modified the text.
- Discussion – What does the second paragraph contribute? Next, how does the third paragraph link to your findings? In addition, how do your results change when using the cut-off from the meta-analysis (reference [30]), instead of your 75th percentile? Finally, you need to highlight again how your work is different from previous studies? What does it add?
Answers: we have conducted the same analysis of the paper you have cited, nevertheless the 103.3 as cut-off was not appropriate for our sample of patients. These types of patients are often not enrolled in clinical studies because they are polypathological and in advanced disease; what has emerged from our study is new data on this category of patients
- English needs to be checked and improved across the manuscript.
Answer: we checked English with a mother tongue writer.
All in all, thank you for your precious suggestions.
Round 2
Reviewer 1 Report
The authors have made many changes to the article. Regarding my previous considerations, they have clarified that the NYHA functional class 4 is not the baseline but at the time of admission, they have complemented the information on the diagnosis of HF, they have made the necessary modifications in Table 1 and in the methods section etc. I want to congratulate them because they have really improved the article. I think that after these clarifications the article provides interesting information for readers, although I would like some aspects to be addressed:
- The most important aspect is that in Table 1 I find clear differences between the 2 groups in several variables: Left bundle branch block (13% vs 25%), pacemaker-ICD (19% vs 36%), B-blockers (58% vs 86%), amiodarone (1% vs 14%), however in all of them p-value is ns?? I think that the statistical analysis in Table 1 should be reviewed since there seems to be significant differences in variables with impact on prognosis and/or repolarization that could act as confounders and some of them should be included in the multivariate analysis, especially amiodarone.
Minor changes:
- Please include the aim of the study at the end of the introduction section.
- Please include the department in which the patients were admitted (at the beginning of the methods section).
- Please, include previous NYHA functional class of the participants (if this information is available)
- Table 1: Add a table foot where you define the different acronyms. Also define CHF with depressed systolic function (LVEF <50%?) and CHF with preserved systolic function.
- Please, move the limitations section to the end of the discussion section and add that this is an observational single-center study, and more studies will be necessary to confirm the prognostic value of repolarization in heart failure management and its usefulness in telemedicine.
Author Response
#Reply to Reviewer 1
we would like to thank Reviewer 1 not only for having appreciated the revision that we have carried out, but also for having suggested a further change that brought an undoubted value to the results of the study. According to the major changes required by the reviewer 2, you will probably find many revised paragraphs in the new version. Nevertheless, we have done our best to put and maintain your previous observations.
Answer point by point:
“… I want to congratulate them because they have really improved the article. I think that after these clarifications the article provides interesting information for readers, although I would like some aspects to be addressed:
- The most important aspect is that in Table 1 I find clear differences between the 2 groups in several variables: Left bundle branch block (13% vs 25%), pacemaker-ICD (19% vs 36%), B-blockers (58% vs 86%), amiodarone (1% vs 14%), however in all of them p-value is ns?? I think that the statistical analysis in Table 1 should be reviewed since there seems to be significant differences in variables with impact on prognosis and/or repolarization that could act as confounders and some of them should be included in the multivariate analysis, especially amiodarone.”
Answer: Your invaluable observation on amiodarone has prompted us to completely revise the statistical analysis concerning the variables of table 1. In fact, precisely with regard to amiodarone we had made a terrible mistake, which we apologize for. We have entered the modified p-value in table 1. Furthermore, we have redone the Cox regression in which it is confirmed as a no significant variable, probably due to the small number of subjects taking this drug.
|
|
Group 1 |
Group 2 |
|
|
|
|
N: 85 |
N: 28 |
|
|
|
|
|
|
Χ2 |
p |
|
Left Bundle Branch Block, n (%) |
11(13) |
7(25) |
1.579 |
0.208 |
|
|
|
|
|
|
|
Pacemaker- ICD, n (%) |
16(19) |
10(36) |
1.992 |
0.156 |
|
β-blockers, n (%) |
49(58) |
24(86) |
1.444 |
0.229 |
|
Amiodarone, n (%) |
1(1) |
4(14) |
7.387 |
0.006 |
- Minor changes:
Answer: we have followed all your precious suggestions for minor changes.
Finally, we really are in debt with you.

Reviewer 2 Report
Unfortunately authors have not incorporated most of the suggestions I had indicated in my first round. As an example, the objectives of the study are not clearly defined, and the introduction is still not well motivated.
Author Response
#Reply to Reviewer 2
2nd Round
General considerations:
Unfortunately, authors have not incorporated most of the suggestions I had indicated in my first round. As an example, the objectives of the study are not clearly defined, and the introduction is still not well motivated.
Answer: we really apologize with You and, in the present version, we tried again to accomplish the suggested improvements. Specifically, we rewrote completely the introduction to improve the its comprehensibility and fluency as well as to better clarify the aim of the study. In addition, we have added, following your previous suggestion, another figure comparing ECG data between died versus survived patients. To avoid a figures’ overload, we merged the Te Kaplan Meier survival curves, thus clarifying that the Te predict not only the cardiovascular mortality but also the all-causes one. This a very important point for us, because the remote-controlled follow up by the means of ECG devices could became an important tool even for general practitioners.
Eventually, we revised accurately the English language trying to correct all typos as well as to reword some convoluted/misleading sentences to make the final message more comprehensible. Notwithstanding we believe that the present version is significantly improved, should we have failed to address some points correctly we apologize, and remain open to suggestions for further changes.